# Identify Drug-Resistant Pathogens in Patients with Community-Acquired Pneumonia

**Francesco Amati** [1,2,*], **Francesco Bindo** [2,3], **Anna Stainer** [1,2], **Andrea Gramegna** [3,4], **Marco Mantero** [3,4], **Mattia Nigro** [1,2], **Linda Bussini** [1,5], **Michele Bartoletti** [1,5], **Francesco Blasi** [3,4] and **Stefano Aliberti** [1,2]

1   Department of Biomedical Sciences, Humanitas University, Via Rita Levi Montalcini 4, Pieve Emanuele, 20072 Milan, Italy
2   Respiratory Unit, IRCCS Humanitas Research Hospital, Via Manzoni 56, Rozzano, 20089 Milan, Italy
3   Department of Pathophysiology and Transplantation, Università degli Studi di Milano, 20122 Milan, Italy
4   Respiratory Unit and Cystic Fibrosis Adult Center, Fondazione IRCCS Ca' Granda Ospedale Maggiore Policlinico, 20122 Milan, Italy
5   Infectious Diseases Unit, IRCCS Humanitas Research Hospital, Via Manzoni 56, Rozzano, 20089 Milan, Italy
*   Correspondence: francesco.amati@hunimed.eu; Tel.: +39-02-8224-3042

**Highlights:**

**What are the main findings?**

- The identification of patients with CAP at high risk for resistant pathogens is of outstanding clinical interest due to the worse outcome of these patients.
- However, the HCAP classification and the scores proposed in literature to identify resistant pathogens in CAP are overly sensitive, leading to inappropriately broad-spectrum antibiotic use.

**What is the implication of the main finding?**

- As suggested by the new IDSA/ATS guidelines, it is crucial to generate local data concerning DRP in order to identify and validate risk factors at a local level.
- Identification of new, rapid and specific diagnostic tests for DRP represents a clinical priority to improve the outcomes of CAP patients.

**Abstract:** A substantial increase in broad-spectrum antibiotics as empirical therapy in patients with community-acquired pneumonia (CAP) has occurred over the last 15 years. One of the driving factors leading to that has been some evidence showing an increased incidence of drug-resistant pathogens (DRP) in patients from a community with pneumonia, including methicillin-resistant *Staphylococcus aureus* (MRSA) and *Pseudomonas aeruginosa*. Research has been published attempting to identify DRP in CAP through the implementation of probabilistic approaches in clinical practice. However, recent epidemiological data showed that the incidence of DRP in CAP varies significantly according to local ecology, healthcare systems and countries where the studies were performed. Several studies also questioned whether broad-spectrum antibiotic coverage might improve outcomes in CAP, as it is widely documented that broad-spectrum antibiotics overuse is associated with increased costs, length of hospital stay, drug adverse events and resistance. The aim of this review is to analyze the different approaches used to identify DRP in CAP patients as well as the outcomes and adverse events in patients undergoing broad-spectrum antibiotics.

**Keywords:** community-acquired pneumonia; drug-resistant pathogens; broad-spectrum antibiotics



## 1. Introduction

Pneumonia is one of the most common and life-threatening diseases worldwide [1]. Proper and timely empiric antibiotic treatment is crucial to improve prognosis in patients with community-acquired pneumonia (CAP) [2]. A substantial increase in broad-spectrum

antibiotics as empirical therapy in CAP patients occurred over the last 15 years [3–5]. One of the driving factors leading to that has been some evidence showing an increased incidence of drug-resistant pathogens (DRP) in patients coming from the pneumonia community, including methicillin-resistant *Staphylococcus aureus* (MRSA) and *Pseudomonas aeruginosa* [6–15]. Different experiences have been published attempting to identify DRP in CAP through the implementation of probabilistic approaches in clinical practice [16–27]. However, recent epidemiological data showed that the incidence of DRP in CAP varies significantly according to local ecology, healthcare systems and countries where the studies were performed [1,28]. Several studies also questioned whether broad-spectrum antibiotic coverage might improve outcomes in CAP [29–33], even though it is widely documented that the overuse of broad-spectrum antibiotics is associated with increased costs, length of hospital stay (LOS), drug-related adverse events and microbial resistance [29,31,32,34–36]. The aim of this review is to analyze different approaches used to identify DRP in CAP patients as well as the outcomes and adverse events in patients undergoing broad-spectrum antibiotics.

## 2. The Impact of Broad-Spectrum Antibiotic Use in CAP

The selection of antibiotic therapy in CAP is a challenging and thorny issue. In most cases, the choice is empirical due to the results of the initial microbiological work-up being in progress or unavailable at the time of antibiotic initiation. In selecting the correct empiric antibiotic therapy, clinicians tend to evaluate risk factors and the need to cover DRP that, if not treated properly, can lead to adverse outcomes in patients, including mortality [2,9,11,28,37,38]. The use of the healthcare-associated pneumonia (HCAP) definition to identify patients at risk for DRP led to an over-utilization of broad-spectrum antibiotics, most of which have been proven unnecessary [39–41]. The prevailing practice of doing something extra (such as extending the spectrum of empiric antibiotic therapy) feels more responsive, responsible, and patient-centric. However, the use of broad-spectrum antibiotics should not always be considered the safest (nor wiser) choice for CAP patients. The reason for this is that broad-spectrum antibiotics, as reported in the literature, could have a negative impact on a patient's prognosis and cause potentially harmful effects on public health in terms of spreading antibiotic resistance and consuming healthcare resources [29–36,40] (Figure 1).

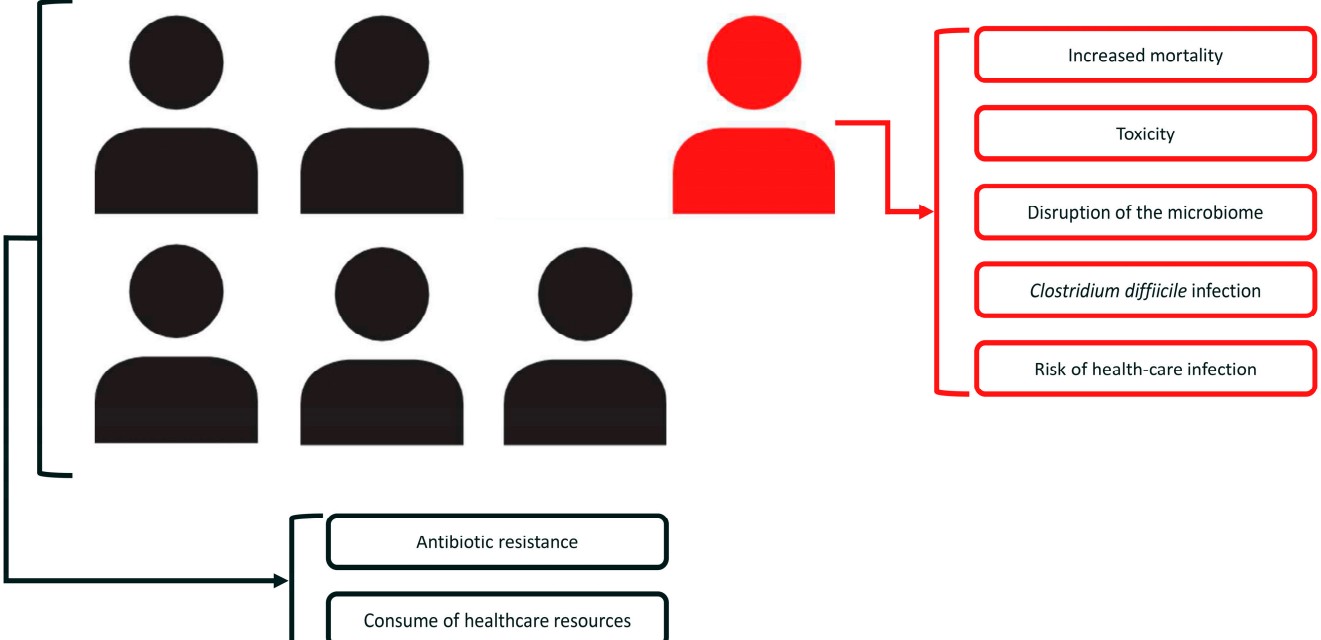

**Figure 1.** Consequences of broad-spectrum antibiotic overuse. In red: negative impact on a single patient's prognosis. In black: potentially harmful effects on public health.

### 2.1. Impact on Individual-Patient Level

Several studies showed that broad-spectrum antibiotics are associated with poor outcomes in CAP, in particular with increased mortality [33,35]. Different aspects related to broad-spectrum antibiotic use contribute to this evidence. First, broad-spectrum antibiotic use is associated with increased LOS, which also increases both the chance of nosocomial infection [34,42–45] and drug-related adverse events [29,34,46–48].

One of the most common broad-spectrum regimens used in hospitalized patients with severe CAP presenting risk factors for PDRs is the combination of piperacillin/tazobactam (because of its activity against both Gram-positive and Gram-negative bacteria, and in particular, *P. aeruginosa*) and vancomycin (commonly selected for its activity against MRSA) [20]. However, vancomycin has been shown to be inferior to linezolid for the treatment of MRSA nosocomial pneumonia [49]. Notably, the combination of vancomycin and piperacillin/tazobactam is associated with a significant risk of nephrotoxicity [47,48]. Vancomycin monotherapy is also associated with nephrotoxicity, although the risk of acute kidney injury (AKI) is lower compared to the above-mentioned combination therapy [50]. In a meta-analysis including 24,799 patients, the rate of AKI is 22% in patients treated with piperacillin/tazobactam *plus* vancomycin compared to less than 13% in patients treated with vancomycin alone or in combination with other beta-lactams [47]. Several reasons might explain the increased risk of AKI due to the vancomycin-piperacillin/tazobactam combination therapy compared to vancomycin therapy alone: (1) nephrotoxicity due to vancomycin could be linked to its accumulation in proximal tubules resulting in acute tubular necrosis [51]; (2) vancomycin causes oxidative stress [52]; (3) semisynthetic penicillins, such as piperacillin/tazobactam, cause acute interstitial nephritis [53]; (4) piperacillin/tazobactam decreases the clearance of vancomycin leading to vancomycin accumulation, subsequently increasing AKI risk [54]. The development of AKI in pneumonia is associated with major adverse kidney events, such as chronic kidney disease and dialysis, and it also increases mortality rates [55].

AKI is the most common adverse event related to the use of broad-spectrum antibiotics, but other adverse effects are widely reported, including cytopenia, encephalopathy, hypersensitivity, and gastrointestinal side effects [56–59]. Patients exposed to broad-spectrum antibiotics show an increased risk of *Clostridioides difficile* infection (CDI), which is 4-fold greater in CAP patients [60]. Alterations in the gut microbiome related to the use of broad-spectrum antibiotics contribute to the acquisition of CDI [61]. Furthermore, the use of broad-spectrum antibiotics also affects the lung microbiome, potentially leading to an increased risk of subsequent infections and readmission for infection-related issues [62].

Patients with pneumonia and CDI infection have an increased LOS and a higher rate of in-hospital mortality compared to those without CDI [63]. Further studies are needed to better explore the complex interaction between the lung microbiome and the use of broad-spectrum antibiotics.

### 2.2. Impact on Public Health

The most widely recognized consequence of broad-spectrum antibiotic use at the community level is the risk of resistance which perpetuates a vicious cycle leading to an increased risk of infection with a resistant pathogen in the community [64]. The consumption of healthcare resources also represents a crucial issue. As described in the previous section, broad-spectrum antibiotics are associated with prolonged LOS [34,42–45]. Furthermore, antibiotics account for approximately 20% of all drug-related emergency department visits in the United States of America (USA), and nearly 80% of these visits are attributable to presumptive allergic reactions [58]. This implies a massive use of human resources (e.g., nurses, doctors), therapies, and diagnostic tools that ultimately result in increased healthcare costs [58,65]. Moreover, there is a striking paucity of new drugs active against multidrug-resistant bacteria [66–70]. Finally, the massive use of antibiotics during the severe acute respiratory syndrome coronavirus 2 (SARS-CoV-2) pandemic, especially those with a broad antibacterial spectrum, might have hindered the progress and the results

achieved in recent years by international research in this field [70]. Compared to the number of hospitalized patients with Coronavirus disease 2019 (COVID-19) who received antibiotics, far fewer patients admitted for COVID-19 had common bacterial infections. Only 20% of those admitted with SARS-CoV-2 infection were diagnosed with suspected or confirmed bacterial pneumonia, and less than 5% were diagnosed with a community-acquired infection [71]. Eventually, the overuse of broad-spectrum antibiotics in the context of few innovative or new antibiotics in the drug development pipeline may frustrate the advances and efforts in antibiotic research leading to resistance to new antibiotics.

## 3. DRP

### 3.1. Definition

The definition of DRP in CAP changed significantly over the last 15 years. Considering different studies that evaluated the presence of DRP in CAP, substantial differences exist in defining these pathogens [18,22,24,25,72–74]. *Park* et al. included MRSA, *P. aeruginosa*, *Acinetobacter baumannii*, *Stenotrophomonas maltophilia*, and extended-spectrum beta-lactamase (ESBL)+ *Enterobacteriaceae* as DRP [50] based on previous reports showing poor clinical outcomes in patients with ventilator-associated pneumonia who were infected with these microorganisms [72]. However, pathogens that are potentially susceptible to antibiotics commonly used for CAP (e.g., *P. aeruginosa* could be susceptible to third-generation cephalosporin or respiratory fluoroquinolones) are included in this definition. Prina et al. used the acronym "PES" (Pseudomonas aeruginosa, Enterobacteriaceae extended-spectrum β-lactamase-positive, and methicillin-resistant Staphylococcus aureus) to identify pathogens not covered by the initial empiric treatment for CAP suggested by guidelines such as *P. aeruginosa*, ESBL + *Enterobacteriaceae*, and MRSA [24]. Recently an international panel of experts proposed a standard definition for multi-drug resistance (MDR) pathogens according to the results of susceptibility tests [73]. Shido and Falcone used this standard definition to characterize the population of multi-drug resistance pathogens in their cohorts of patients [22,25]. The definition of DRP, according to the susceptibility tests, can potentially reduce unnecessary use of broad-spectrum antibiotics.

### 3.2. Prevalence

Important differences exist in the prevalence of DRP and the prevalence of each single DRP across different clinical studies worldwide (Table 1) [18,19,21–27,74]. MRSA and *P. aeruginosa* are the most frequently isolated DRP. Studies in culture positive patients that were performed in the USA showed a higher prevalence of DRP, and in particular MRSA, compared to the rest of the world [19,26,74]. European studies showed a lower prevalence of DRP, and the rate of *P. aeruginosa* or MRSA in CAP seemed to be lower than 6% [21,24,25]. These differences across countries and continents could be explained by both local ecologies and the denominator used across the different studies. For example, accessibility and characteristics of long-term care facilities (LTFCs) and nursing homes (NHs) vary greatly among different countries. Studies from the USA found different microbiological patterns in patients with CAP coming from NH or LTFC in comparison to studies performed in Europe. In addition, while an association between DRP and nursing home-acquired pneumonia (NHAP) has been observed in the USA, this association has not been confirmed in Europe. [19,21,24–26,74].

**Table 1.** Numbers of DRP identified in studies using clinical prediction models for DRP detection in CAP [18,19,21–27,74].

| First Author and Year | Country | Number of Patients | Culture Positive | Number of DRP (%) | MRSA (%) | *Pseudomonas aeruginosa* (%) | Other DRP (%) |
|---|---|---|---|---|---|---|---|
| Shorr 2008 [26] | USA | 639 | 639 (100%) | 289 (45.2%) | 157 (24.6%) | 120 (18.8%) | 47 (7.4%) |
| Schrieber 2010 [74] | USA | 190 | 190 (100%) | 62 (32.6%) | 35 (18.4%) | 25 (13.2%) | 2 (1%) |
| Aliberti 2012 [21] | Italy | 935 | 170 (18%) | 33 (3.5%) | 16 (1.7%) | 7 (0.7%) | 10 (1.1%) |
| Park 2013 [18] | South Korea | 339 | 339 (100%) | 122 (36%) | 27 (8%) | 58 (17.1%) | 37 (10.9%) |
| Shindo 2013 [22] | Japan | 1413 | 795 (56,3%) | 170 (12.3%) | 77 (5.4%) | 79 (5.6%) | 14 (1%) |
| Ma 2014 [23] | China | 450 | 450 (100%) | 69 (15.3%) | 8 (1.8%) | 56 (12.4%) | 6 (1.3%) |
| Prina 2015 [24] | Spain | 1597 | 1597 (100%) | 108 (6.8%) | 21 (1.3%) | 72 (4.5%) | 15 (0.9%) |
| Falcone 2015 [25] | Italy | 900 | 300 (33.3%) | 99 (11%) | 50 (5.6%) | 17 (1.9%) | 32 (3.5%) |
| Webb 2016 [19] | USA | 400 | 400 (100%) | 124 (31%) | 57 (14.2%) | 34 (8.5%) | 33 (8.2%) |
| Rothberg 2022 [27] | USA | 138,940 | 12,181 (8.8%) | 5200 (3.8%) | Not analyzed | Not analyzed | Not analyzed |

Abbreviations: USA: United States of America; DRP: drug-resistant pathogens; MRSA: Methicillin-resistant *Staphylococcus aureus*; Other DRP: drug-resistant pathogens that are not Methicillin-resistant *Staphylococcus aureus* or *Pseudomonas aeruginosa*.

### 3.3. The Identification of DRP: The Failing of the HCAP Classification

In 2005 the IDSA/ATS guidelines introduced the concept of healthcare-associated pneumonia (HCAP) with the aim of predicting DRP in patients with CAP [20]. HCAP was defined by the presence of at least one among the following risk factors: (1) residence in a NH or extended-care facility, (2) home infusion therapy, (3) antimicrobial therapy in the preceding 90 days, (4) chronic dialysis in the preceding 30 days, (5) home wound care or (6) a family member infected or colonized by a MDR pathogen. According to those guidelines, patients with one of these criteria should be treated with broad-spectrum antibiotics, including two antipseudomonal drugs and one anti-MRSA agent, if a risk factor for MRSA was present [20]. The idea of HCAP arose from single-center, retrospective data and, mainly, expert opinions [16,74,75]. Subsequent studies, in particular those conducted in Europe and with a prospective design, showed a different situation [15,21]. A meta-analysis published in 2013 by Chalmers et al. evaluated the accuracy of the HCAP classification in identifying patients with CAP due to MDR pathogens [39]. The authors found that the HCAP definition is neither sensitive nor specific in identifying patients at risk for MDR bacteria, especially in studies of high quality and perspective. Furthermore, the authors found a significant increase in the use of anti-pseudomonal and anti-MRSA agents as empirical treatment in the setting of CAP since the classification of HCAP was introduced without any apparent improvement in patients' outcomes [76]. According to this evidence, the new ATS/IDSA guidelines recommended abandoning the HCAP classification [77].

### 3.4. The Identification of DRP: The Surge of Probabilistic Approaches

In order to go beyond the limitations of the HCAP definition, multiple clinical prediction models using a probabilistic approach for DRP in CAP emerged in the last 15 years (Table 2) [18,19,21–27,74]. All these scores showed better AUROC curves compared to the HCAP definition in identifying CAP patients due to DRP. However, some considerations should be taken into account when different scores are analyzed. First of all, as previously described, a substantial difference exists in the prevalence of DRP, and heterogeneous definitions of DRP are used across different studies [18,19,21–27,74]. Second, the definition of each risk factor varies across different studies. For instance, the temporal definition of recent intravenous antibiotic use ranges from 30 days to 90 days [18,22,23]. Likewise, the time limit to define "recent" hospitalization is 90 days in the majority of papers, although some papers show an increased risk for resistant bacteria up to 1 year [18,19,21–27,74]. Third, the length of previous hospital stays is not always considered, as well as the setting in which the patients are hospitalized. Indeed, patients admitted to an intensive care unit

(ICU) seem to be exposed to a different ecology compared to patients admitted to the general ward. Fourth, the definition of chronic kidney disease (CKD) is widely heterogeneous: Aliberti et al. defined CKD if the patient had a level of blood creatinine > 1.2 mg/dL, while Prina et al. defined CKD if there was a history of decreased kidney function (defined as a glomerular filtration rate lower than 60 mL/min/1.73 m$^2$) of three or more months [21,24]. Fifth, the inclusion criteria in each study were highly heterogeneous. The importance of including all consecutive CAP patients and not only those with a culture-positive result -the so-called "denominator issue"- has been previously discussed. Immunosuppression is considered an inclusion criterium and also a risk factor in some of these papers [21,23,25,74]. However, an important amount of evidence showed that these patients should be considered as a separate entity, and international guidelines do not address the management of CAP patients who are immunocompromised [77,78]. Sixth, validation cohorts are essential to confirm the robustness of a score. The population analyzed by a score is different in terms of comorbidities, age and/or setting in which the score was developed and tested (ICU versus general ward versus outpatient setting). As an example, the Ma cohort consists of elderly patients with a mean age of 80 years, and the generalizability of these data in a different cohort is a thorny issue [23]. Last but not least, these scores are able to identify patients at risk for a CAP due to DRP in general, but they are neither developed nor validated to identify the risk for a specific DRP (such as MRSA or *Pseudomonas* or ESBL+). A risk factor for a DRP is not necessarily a risk factor for another DRP. The literature shows several risk factors for *Pseudomonas* and MRSA but not all these risk factors overlap (Table 3) [6–11,18,19,21–26,74]. Furthermore, treatment for Gram-positive bacteria, such as MRSA, involves the use of antibiotics that are ineffective against resistant Gram-negative bacteria, such as *P. aeruginosa*, and vice versa. According to these considerations, the use of these scores might also lead to antibiotic overuse.

**Table 2.** Risk scores for DRP in CAP derived using a probabilistic approach in studies published in the last 15 years [18,19,21–27,74]. Points in the different models are related to the ODDs ratio.

| First Author and Year | Country | DRP | Number of Patients | Risk Factors and Points | Design | External Validation | Threshold for Definition of Risk for DRP |
|---|---|---|---|---|---|---|---|
| Shorr 2008 [26] | USA | MRSA, *P aeruginosa*, extended-spectrum β-lactamase–producing *Klebsiella* species, and other nonfermenting gram-negative bacteria * | 639 | Recent hospitalization = 3<br>Nursing home residence = 2<br>Hemodialysis = 2<br>ICU admission= 1 | Retrospective Single center Culture positive Hospitalized | Yes | ≥1 point |
| Schrieber 2010 [74] | USA | MRSA, *P. aeruginosa*, ESBL-producing bacteria | 190 | Immunosuppression = 3<br>Admission from long-term care= 2<br>Prior antibiotics, 1 | Retrospective Single center Culture-positive ICU patients | No | ≥2 points |
| Aliberti 2012 [21] | Italy | MRSA; *P. aeruginosa* resistant to antipseudomonal penicillins, cephalosporins, carbapenems, and quinolones; *Stenotrophomonas maltophilia*; vancomycin-resistant *Enterococcus*; *A. baumanii*; ESBL–producing *Enterobacteriaceae*; other nonfermenting gram-negative bacilli | 935 | Chronic renal failure =5<br>Hospitalization in the preceding 90 days =4<br>Residence in a nursing home =3<br>Others (cerebrovascular disease, diabetes, COPD, immunosuppression, home wound care, prior antimicrobial therapy and home infusion therapy) = 0.5 | Prospective Single center Ward and ICU All CAP patients | Yes | ≥3 points |
| Park 2013 [18] | South Korea | MRSA, *P.aeruginosa*, *A. baumannii*, *S. maltophilia*, and ESBL-producing *Enterobacteriaceae* | 339 | Tube feeding = 5<br>Recent hospitalization= 3<br>Recent (30 days) intravenous antibiotics =2<br>Admission from long-term care facility= 1<br>Recent (30 days) chemotherapy =1<br>Recent (30 days) wound care = 1<br>Chronic dialysis =1 | Retrospective Single center Ward and ICU Culture-positive | No | ≥3 points |
| Shindo 2013 [22] | Japan | Any microorganism resistant to at least one agent in three or more groups of antibiotics | 1413 | Recent hospitalization (last 90 days) =1<br>Immunosuppression =1<br>Home infusion therapy (last 90 days) =1<br>Use of gastric acid-suppressive agents =1<br>Tube feeding =1<br>Non-ambulatory status =1 | Prospective Multicenter Inpatients All CAP patients | Yes | ≥3 points |
| Ma 2014 [23] | China | MRSA, *P. aeruginosa*, extended-spectrum β-lactamase (ESBL)-producing *Enterobacteriaceae* and *A. baumannii*. | 450 | Bronchiectasis =14<br>Recent hospitalization = 5<br>Severe pneumonia = 2<br>Others (nursing home residence, home infusion therapy, chronic wound care, chronic dialysis or immunosuppression) = 0.5 each | Prospective Single center Inpatients Culture positive | No | ≥2.5 points |
| Prina 2015 [24] | Spain | *P. aeruginosa*, ESBL-positive *Enterobacteriaceae*, and MRSA | 1597 | Age 40–65 years =1<br>Age >65 years =2<br>Male =1<br>Previous antibiotic use =2<br>Chronic respiratory disease (COPD or bronchiectasis) = 2<br>Chronic renal disease =3<br>Consciousness impairment= 2<br>Fever = 1 | Prospective Single center Inpatients Culture positive | Yes | ≥2 points |

**Table 2.** *Cont.*

| First Author and Year | Country | DRP | Number of Patients | Risk Factors and Points | Design | External Validation | Threshold for Definition of Risk for DRP |
|---|---|---|---|---|---|---|---|
| Falcone 2015 [25] | Italy | MRSA, *S. maltophilia*, ESBL–producing or carbapenem-resistant *Enterobacteriaceae*, PLUS any bacterial strain non-susceptible to at least one agent in three or more antimicrobial categories. | 900 | HCAP criteria= 1<br>Bilateral pulmonary infiltrations= 0.5<br>Pleural effusion= 0.5 $PaO_2/FiO_2$ <300 = 1.5 | Prospective Single center All CAP patients | Yes | ≥3 points |
| Webb 2016 [19] | USA | MRSA, *P. aeruginosa*, *Enterobacteriaceae* drug-resistant | 400 | Prior antibiotics = 2<br>Residence in a long-term care facility = 2<br>Tube feeding = 2<br>Infection with a drug-resistant pathogen in the previous year = 2 Hospitalization (60 days) = 1<br>Chronic pulmonary disease= 1<br>Poor functional status= 1<br>Gastric acid suppression = 1<br>Wound care = 1<br>MRSA colonization in the previous year = 1<br>Resistant organism in previous year [†] = 2.5<br>Invasive mechanical ventilation (IMV) = 2 | Retrospective Multicenter Culture positive | Yes | ≥4 points |
| Rothberg 2022 [27] | USA | Any organism resistant to either a quinolone or the combination of a third-generation cephalosporin and a macrolide | 138,940 | Pressure ulcer = 1.5<br>Vasopressor Administration = 1.5<br>Paralysis = 1.5<br>Admission to intensive care unit (ICU) = 1.5<br>Low functional status/weight loss = 1.5<br>Hospital admission in previous year = 1.5<br>Admitted from skilled nursing or intermediate care Facility = 1.5<br>Chronic pulmonary disease = 1.5<br>Male sex = 1.5<br>Current tobacco smoker = 1 | Retrospective Multicenter Inpatients All CAP patients | No | >4 points |

\* *Stenotrophomonas maltophilia*, *Burkholderia cepacia*, *Sphingomonas paucimobilis*, *Achromobacter xylosoxidans* and *Acinetobacter baumanii*; † Resistant either to a third-generation cephalosporin, ampicillin, or ertapenem, and a macrolide or to a fluoroquinolone; Abbreviations: DRP: drug-resistant pathogens; USA: United States of America; MRSA: Methicillin-resistant *Staphylococcus aureus*; ESBL: extended-spectrum β-lactamase; ICU: intensive care unit; CAP: community-acquired pneumonia; HCAP: healthcare-associated pneumonia.

**Table 3.** Specific risk factors for *MRSA* or *P. aeruginosa* CAP [6–11,18,19,21–26,74].

| Risk Factor | MRSA | P. aeruginosa |
|---|:---:|:---:|
| Comorbidity | | |
| Chronic lung diseases (defined as COPD and or bronchiectasis) | X | X |
| Cerebrovascular diseases | X | |
| Diabetes mellitus | X | |
| Altered mental status | X | |
| Recurrent skin infection | X | |
| Prior exposure | | |
| Prior infection or colonization | X | X |
| Prior antibiotic | X | X |
| Prior hospitalization (12 months) | X | X |
| Prior tracheostomy | | X |
| Demographic characteristics | | |
| Age (<30 years or >79 years) | X | |
| Male gender | | X |
| Enteral tube feeding | X | X |
| Residence in a nursing home | X | |
| Tobacco use | X | |
| Severity of illness | | |
| Severe CAP | X | |
| $PaO_2/FiO_2$ <200 | | X |
| Invasive respiratory or vasopressors support | | X |
| High serum levels of CRP | | X |
| PSI IV or V | X | X |

Abbreviations: MRSA: Methicillin-resistant *Staphylococcus aureus*; CAP: community-acquired pneumonia; COPD: chronic obstructive pulmonary disease; CRP: C-reactive protein; PSI: pneumonia severity index.

## 4. The New IDSA/ATS Guidelines Criteria to Identify DRP

The two most frequent DRP in CAP are MRSA and *P. aeruginosa,* and the latest international guidelines clearly focus their attention on these two pathogens [77]. However, most of the individual risk factors are weakly associated with these pathogens, and no validated scoring systems exist to identify patients with either MRSA or *P. aeruginosa* with sufficiently high positive predictive value to determine the need for empiric extended-spectrum antibiotic treatment (Table 3) [6–11,18,19,21–27,74]. The guidelines recognize that the most consistently strong individual risk factors for respiratory infection due to MRSA or *P. aeruginosa* are the prior isolation of these organisms, especially from the respiratory tract, and/or recent hospitalization and exposure to parenteral antibiotics [77]. The guidelines suggest the use of local prevalence data and locally validated risk factors for MRSA and *P. aeruginosa* [77]. Moreover, in the absence of local data, guidelines recognize that a prior identification of MRSA or *P. aeruginosa* in the respiratory tract predicts a very high risk for these pathogens being the cause of CAP. Therefore, these were sufficient indications to recommend blood and sputum cultures and empiric therapy for these pathogens in patients with CAP in addition to coverage for standard CAP pathogens, with de-escalation at 48 h if cultures are negative. In patients with recent hospitalization and exposure to parenteral antibiotics, guidelines recommend microbiological testing without empiric extended-spectrum therapy for the treatment of non-severe CAP and microbiological testing with extended-spectrum empiric therapy in addition to coverage for standard CAP

pathogens for treatment of severe CAP with de-escalation at 48 h if cultures are negative and the patient is improving (Figure 2). Future studies are needed to validate the criteria proposed by the ATS/IDSA guidelines in order to understand if they are useful to identify CAP patients with MRSA or *P. aeruginosa* without the risk of over or undertreating patients.

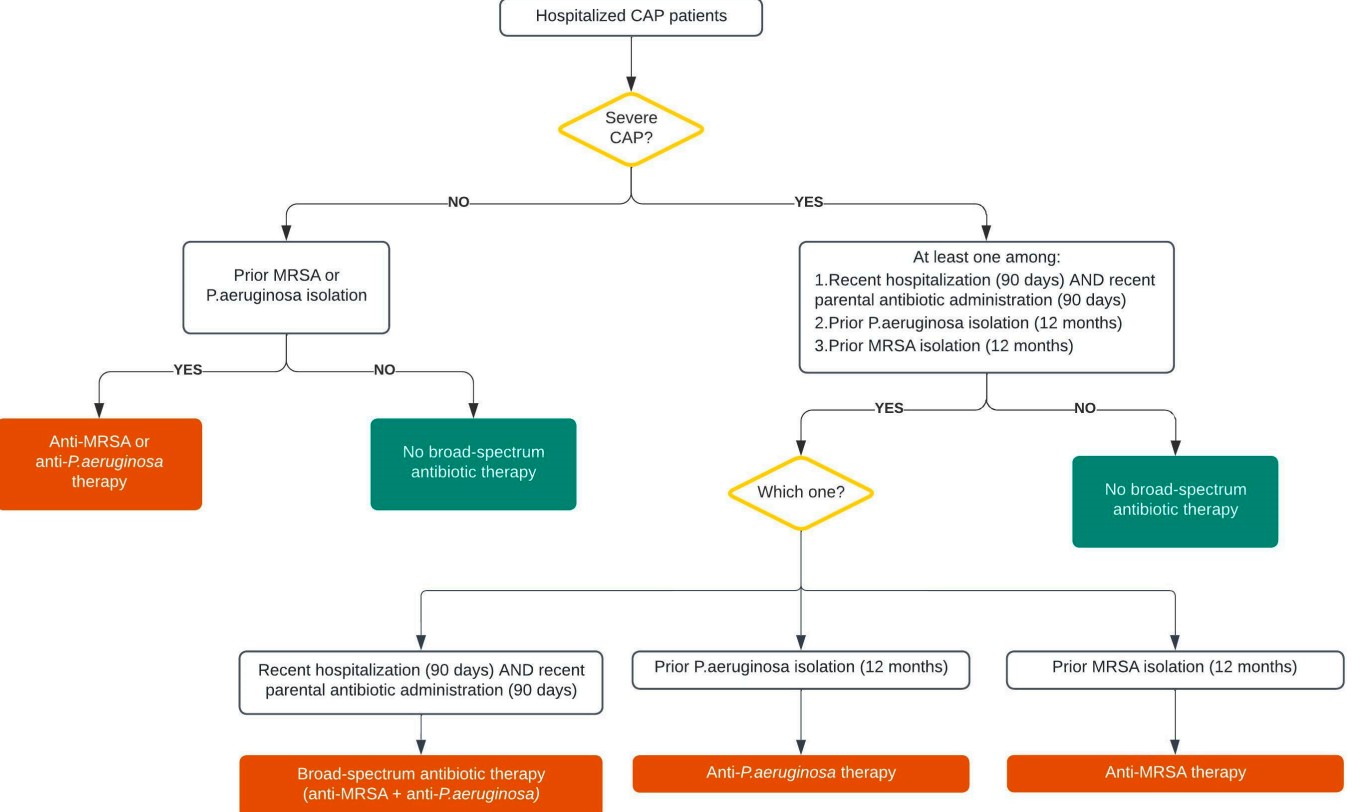

**Figure 2.** Guidelines algorithm for DRP treatment [77].

## 5. Rapid Tests for DRP

The emerging and refining of new diagnostic techniques can help to better identify DRP. Although these methods are available only in a few centers, particularly in the USA, their use may prospectively change the approach used to identify DRP in CAP patients. Molecular methods for nasopharyngeal swabs, sputum and bronchoalveolar lavage (BAL), such as the BioFire Film Array 2.0 Pneumonia Panel, are available in many hospitals in the USA and Europe [75]. Molecular methods are able to quickly (in 2 h or less) identify specific resistance genes in several species of bacteria, including MRSA, *P. aeruginosa* and ESBL+ *Enterobacteriaceae* [79,80]. However, large validation of these methods in pneumonia and, in particular, in CAP is needed in both immunocompetent and immunocompromised patients. A diagnostic tool useful in the suspicion or screening for MRSA pneumonia is the MRSA nasal swab PCR assay [81,82], although *S. aureus,* including MRSA, is a common colonizer of the nares [83]. The absence of MRSA nares colonization has been reported to be a negative predictor of MRSA pneumonia [81]. The results of a recent systematic review showed that nares screening for MRSA had a high specificity and a high negative predictive value for ruling out MRSA pneumonia, particularly in cases of CAP [82]. Indeed, MRSA nares screening represents a valuable tool to streamline empiric antibiotic therapy, especially among patients with non-severe pneumonia who are not colonized with MRSA [81]. However, the positive predictive value is low; therefore, the antibiotic coverage for MRSA in CAP patients with a positive nasal swab is a debated matter, and CAP severity and local prevalence of MRSA as a pathogen should be considered.

## 6. Conclusions

One of the main controversial fields in CAP management is the empiric treatment of patients with potential DRP. The two weights of the balance are represented by the consequences of missing a DRP from one side and antibiotic stewardship/adverse events/ occurrence of resistance from the other side. The question is amplified by the fact that the decision on the empirical antibiotic is usually made before the results of the microbiological work-up are available. There is no doubt that patients with DRP might have worse outcomes, including mortality. However, the inappropriate use of broad-spectrum antibiotics (overuse) has direct consequences not only on the single patient but also at a community level. Furthermore, the incidence of DRP in CAP patients varies considerably according to local ecology. The new guidelines suggest generating local data concerning DRP in order to truly understand the prevalence of DRP across different hospitals and identify and validate risk factors at a local level. This is crucial for antibiotic stewardship because there is no rationale in extending the spectrum of antibiotics as empiric therapy if DRP are uncommon in a specific region or local area. Furthermore, the individual risk factors are weakly associated with a specific DRP, except for prior isolation of these organisms, recent hospitalization and exposure to parenteral antibiotics. The new guidelines suggest the use of broad-spectrum antibiotics in case these risk factors are present and depending on the setting in which the patient is hospitalized (ICU versus ward). In Table 4, we summarize clinical and research priorities concerning DRP management in CAP patients. Future studies are needed to validate the criteria proposed by the ATS/IDSA guidelines in order to understand if the criteria proposed are able to identify CAP patients with MRSA or *P. aeruginosa* without the risk of over or undertreating them.

**Table 4.** Research and clinical priorities concerning DRP.

| | Outstanding Research and Clinical Priorities |
|---|---|
| 1 | Identification and implementation of antibiotic stewardship strategies at a local level, such as prospective audits with intervention and feedback, clinical pathways, and dedicated multidisciplinary teams. |
| 2 | Collection of data concerning the local prevalence of DRP to find stronger locally validated risk factors. |
| 3 | Validation of ATS/IDSA criteria in case of absence of a local database. |
| 4 | Identification of new, rapid, cost-effective, sensitive, and specific diagnostic tests for DRP. |
| 5 | Implementation of new diagnostic strategy in low-income and middle-income countries. |
| 6 | Identification of non-antibiotic drugs (such as bacteriophages) targeting DRP for effective treatment in vivo. |

Abbreviations: DRP: drug-resistant pathogens.

**Author Contributions:** Conceptualization, Conceptualization, F.A.; methodology, F.A. and S.A.; data curation, F.A. and F.B. (Francesco Bindo); writing F.A. and F.B. (Francesco Bindo); review and editing, A.S., A.G., M.M., F.B. (Francesco Blasi), L.B., M.N., and M.B.; visualization, F.A and F.B. (Francesco Bindo); supervision, F.A. and S.A.; project administration, F.A.. All authors have read and agreed to the published version of the manuscript.

**Funding:** This research received no external funding.

**Informed Consent Statement:** Not applicable.

**Data Availability Statement:** No new data were created or analyzed in this study. Data sharing is not applicable to this article.

**Conflicts of Interest:** The authors declare no conflict of interest.

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
