# Peer review of "Identify Drug-Resistant Pathogens in Patients with Community-Acquired Pneumonia"

_arm, doi:10.3390/arm91030018_

Round 1
Reviewer 1 Report
Dear authors,
I have read your paper with great interest and pleasure. It is well written, covering the issues by considering the problem from all aspects, emphasizing the need for laboratory evidence with clear presentation of guidelines and procedures. In your manuscript, what is most important for me is the emphasis on monitoring the local scenario, i.e. the most common causative agents of CAP as well as the resistance of these strains to empirical used antibacterial drugs.
Minor correction I suggest for Table 2. Risk scores for DRP, table could be better designed
Author Response
General comment: I have read your paper with great interest and pleasure. It is well written, covering the issues by considering the problem from all aspects, emphasizing the need for laboratory evidence with clear presentation of guidelines and procedures. In your manuscript, what is most important for me is the emphasis on monitoring the local scenario, i.e. the most common causative agents of CAP as well as the resistance of these strains to empirical used antibacterial drugs.
Response: We would like to thank the reviewer for her/his nice words.
Comment 1: Minor correction I suggest for Table 2. Risk scores for DRP, table could be better designed.
Response to comment 1: We thank the reviewer for her/his comment. We recognize that this comment is important. For this reason, we modified the
Reviewer 2 Report
I read the review with interest. I have the following critical points. A clear definition of CAP should be provided, as there are some mixes within the manuscript.
1. Introduction: some of the cited references in the introductionh are related to HAP or VAP, not CAP, and some of them old (among the 36 studies included).
2. Line 60-65: Not clear, re-write.
3. Figure 1 title should be stand-alone; indicate that there are some points related to the individual levels and others for the population.
4. Line 72-73L Rewrite
5. LOS abbreviation is not mentioned in full initially.
6. Impact on individual-patient leve section: divide into more than one paragraph
7 "The most 75 common broad-spectrum regimen used in CAP is the combination of vancomycin (com- 76 monly selected for its activity against MRSA), and piperacillin/tazobactam (because of its 77 activity against both Gram-positive and Gram-negative bacteria, and in particular P. aeroginosa)." This sentence is without reference, it is not precise or accurate; based in which country? also, it is not for all CAP patients, as some treated as outpatients etc.
8. Line 87: synergistic risk of AKI: try to change the term synergistic
9. Line 111-112: "As described in the previous paragraph"; please replace the word paragraph with section or previously; because some changes may happen while proofreading..
10. Please, double check all the abbreviations: “PES”? NH? MDR? ".....standard definition of MDR pathogens according to the results of susceptibility test [74]. Shido and Falcone used this standard definition to characterize the population of multi-drug resistance pathogens...." Then after several lines: multi-drug resistance (MDR) pathogens
11. Three comments on Table 1: 1. the title is not informative, need some more details. 2. you have to cite the reference according to the Journal [1], the year added value but here not the author name, depending on the style. 3. What you mean with other DRP: please, mention in the legend.
12. The idea of the points mentioned in Table 2 needs more details.
13. Line 255: what you mean with (2)? different reference style?
14. Although you cited the references in the text above Table 3; Table 3 nneds citation, possibly in the legend to be stand-alone.
15. Figure 2 was not mentioned in the text, unless you mean with (2) mentioned in point 3; please, correct.
16. "Molecular methods are able to quickly (1 hour or less)", please, paraphrase, as some require 1.5 or 2 hours and they are still advantage and fast.
17. The manuscript need more recommendations, knowledge gaps and some mention relating to data from the developing countries.
Author Response
General comment: I read the review with interest. I have the following critical points. A clear definition of CAP should be provided, as there are some mixes within the manuscript.
Response: We would like to thank the reviewer for her/his nice words.
Comment 1: Introduction: some of the cited references in the introduction are related to HAP or VAP, not CAP, and some of them old (among the 36 studies included).
Response to comment 1: We thank the reviewer for her/his comment. We recognize that this comment is important. For this reason, we modified the references according to reviewer’s suggestion. We removed old references related to HAP and VAP and we added references coming from new studies in CAP.
Comment 2: Line 60-65: Not clear, re-write.
Response to comment 2: We would like to thank the reviewer for this comment. According to reviewer’s suggestion, we changed the text as follow: “The prevailing practice of doing something extra (as extending the spectrum of the empiric antibiotic therapy) feels more responsive, responsible, and patient-centric. However, the use of broad-spectrum antibiotic should not be always considered as the safest (nor wiser) choice for CAP patients. Indeed, several evidence have showed that broad-spectrum antibiotics could have a negative impact on a single patient’s prognosis and potentially harmful effects on public health in terms of antibiotic resistances spreading and healthcare resources consuming [29-36,40].”
Comment 3: Figure 1 title should be stand-alone; indicate that there are some points related to the individual levels and others for the population.
Response to comment 3: We would like to thank the reviewer for this comment. We modified the first figure and modified the title as follow: “Figure 1. Consequences of broad-spectrum antibiotic overuse. In red: negative impact on a single patient’s prognosis. In black: potentially harmful effects on public health”
Comment 4: Line 72-73L Rewrite
Response to comment 4: We would like to thank the reviewer for this comment. We modified the text as follow: “Several studies showed that broad-spectrum antibiotics are associated with poor outcomes in CAP, in particular an increased mortality [33,35]. “
Comment 5: LOS abbreviation is not mentioned in full initially.
Response to comment 5: We would like to thank the reviewer for this comment. The abbreviation “length of hospital stay (LOS)” was firstly reported in line 47.
Comment 6: Impact on individual-patient level section: divide into more than one paragraph.
Response to comment 6: We would like to thank the reviewer for this comment According to reviewer’s suggestion we divided in 4 paragraphs.
Comment 7: “The most common broad-spectrum regimens used in CAP is the combination of vancomycin (commonly selected for its activity against MRSA), and piperacillin/tazobactam (because of its activity against both Gram-positive and Gram-negative bacteria, and in particular P. aeroginosa)." This sentence is without reference, it is not precise or accurate; based in which country? also, it is not for all CAP patients, as some treated as outpatients etc.
Response to comment 7: We would like to thank the reviewer for this comment. We modified the text as follow: “One the most common broad-spectrum regimen used in hospitalized patients with severe CAP presenting risk factors for PDRs is the combination of piperacillin/tazobactam (because of its activity against both Gram-positive and Gram-negative bacteria, and in particular P. aeruginosa) and vancomycin (commonly selected for its activity against MRSA).”
Comment 8: Line 87: synergistic risk of AKI: try to change the term synergistic.
Response to comment 8: We would like to thank the reviewer for this comment. We modified the text as follow:” Several reasons might explain the increased risk of AKI due to the vancomycin-piperacillin/tazobactam combination therapy compared to vancomycin therapy alone”
Comment 9: Line 111-112: "As described in the previous paragraph"; please replace the word paragraph with section or previously, because some changes may happen while proofreading.
Response to comment 9: We would like to thank the reviewer for this comment. We changed paragraph with section.
Comment 10: Please, double check all the abbreviations: “PES”? NH? MDR? "... standard definition of MDR pathogens according to the results of susceptibility test [74]. Shido and Falcone used this standard definition to characterize the population of multi-drug resistance pathogens...." Then after several lines: multi-drug resistance (MDR) pathogens
Response to comment 10: We would like to thank the reviewer for this comment. We corrected in the text the abbreviations.
Comment 11: Three comments on Table 1: 1. the title is not informative, need some more details. 2. you have to cite the reference according to the Journal [1], the year added value but here not the author’s name, depending on the style. 3. What you mean with other DRP: please, mention in the legend.
Response to comment 11: We would like to thank the reviewer for this comment. We modified the title of the table according to reviewer’s suggestion as follows “Table 1: Numbers of DRP identified in studies using clinical prediction models for DRP detection in CAP [18,19,21-27,75].” Moreover, we added missing abbreviations in the legend. Finally, we added in parenthesis the reference for each study.
Comment 12: The idea of the points mentioned in Table 2 needs more details.
Response to comment 12: We would like to thank the reviewer for this comment. Points in the different model are related to the ODDs ratio. To clarify this, we modified the title of the table as follows” Table 2. Risk scores for DRP in CAP derived using a probabilistic approach in studies published in the last 15 years [18,19,21-27,75]. Points in the different models are related to the ODDs ratio.”
Comment 13: Line 255: what you mean with (2)? different reference style?
Response to comment 13: We would like to thank the reviewer for this comment. It is related to Figure 2. Thus, we modified the text according to this.
Comment 14: Although you cited the references in the text above Table 3; Table 3 needs citation, possibly in the legend to be stand-alone.
Response to comment 14: We would like to thank the reviewer for this comment. We added the citation to the table. Thus, we modify the title of the table as follows “Specific risk factors for MRSA or P. aeruginosa CAP [6-11,18,19,21-26,75].”
Comment 15: Figure 2 was not mentioned in the text, unless you mean with (2) mentioned in point 3; please, correct.
Response to comment 15: We would like to thank the reviewer for this comment. According to reviewer’s suggestion we modified the text.
Comment 16: “Molecular methods are able to quickly (1 hour or less)", please, paraphrase, as some require 1.5 or 2 hours, and they are still advantage and fast.
Response to comment 16: We would like to thank the reviewer for this comment. We modified the time needed for these methods adding “2 hours or less” in the text. The sentence has been modified as follows “Molecular methods are able to quickly (2 hour or less) identify specific resistance genes in several species of bacteria including MRSA, P. aeruginosa and ESBL+ Enterobacteriaceae [80,81].”
Comment 17: The manuscript needs more recommendations, knowledge gaps and some mention relating to data from the developing countries.
Response to comment 17: We would like to thank the reviewer for this comment. We added a table with research outstanding and research priorities in this field.
|
Outstanding research and clinical priorities |
|
|
1 |
Identification and implementation of antibiotic stewardship strategies at local level such as prospective audit with intervention and feedback, clinical pathways, and dedicated multidisciplinary teams. |
|
2 |
Collection of data concerning the local prevalence of DRP to find stronger locally validated risk factors. |
|
3 |
Validation of ATS/IDSA criteria in case of absence of a local database. |
|
4 |
Identification of new, rapid, cost-effective, sensitive, and specific diagnostic tests for DRP. |
|
5 |
Implementation of new diagnostic strategy in low-income and middle-income countries. |
|
6 |
Identification of non-antibiotics drugs (such as bacteriophages) targeting DRP for effective treatment in vivo. |
Reviewer 3 Report
Dear Authors, thank you for giving me the opportunity to review your manuscript. Please find some comments below:
4. What specific improvements should the authors consider? Evolution of empirical antimicrobial therapy in guidelines for treatment of CAP 5. Are the conclusions consistent with the evidence and arguments presented and do they address the main question posed? Yes 6. Are the references appropriate? Yes 7. Please include any additional comments on the tables and figures. None
Minor comments:
Line 21,41: please change the wording “through the implementation of approaches in clinical practice
Line 30,35: remove and add spaces where it is necessary
Line 56: delete pathogen after DRP abbreviation
Line 166: DRP
Line 228-229: Please, italicize the microorganisms
Line 255: Please, comment on the (2)
Author Response
General comment: It is the structured review about empirical therapy of community-acquired pneumonia (CAP) and antimicrobial resistance (AMR) of causative pathogens The topic is relevant to the advances in respiratory medicine journal summarizing trends of increased consumption of broad-spectrum antibiotics due to the empirical antimicrobial therapy of CAP. The manuscript raises the need to assess ATS/IDSA guidelines criteria to identify CAP patients with drug-resistant pathogens.
Response: We would like to thank the reviewer for her/his nice words.
Minor comments:
Comment 1: Line 21,41: please change the wording “through the implementation of approaches in clinical practice”.
Response to comment 1: We thank the reviewer for her/his comment. We recognize that this comment is important. For this reason, we modified the text according to reviewer’s suggestion.
Comment 2: Line 30,35: remove and add spaces where it is necessary.
Response to comment 2: We thank the reviewer for her/his comment. We recognize that this comment is important. For this reason, we modified the text according to reviewer’s suggestion.
Comment 3: Line 56: delete pathogen after DRP abbreviation.
Response to comment 3: We thank the reviewer for her/his comment. We recognize that this comment is important. For this reason, we modified the text according to reviewer’s suggestion.
Comment 4: Line 166: DRP.
Response to comment 4: We thank the reviewer for her/his comment. We recognize that this comment is important. For this reason, we modified the text according to reviewer’s suggestion.
Comment 5: Line 228-229: Please, italicize the microorganisms.
Response to comment 5: We thank the reviewer for her/his comment. We recognize that this comment is important. For this reason, we modified the text according to reviewer’s suggestion.
Comment 6: Line 255: Please, comment on the (2).
Response to comment 6: We thank the reviewer for her/his comment. We recognize that this comment is important. For this reason, we modified the text according to reviewer’s suggestion adding “Figure” before the 2.
Round 2
Reviewer 2 Report
Comments are addressed
Reviewer 3 Report
Thank you for your comments and revisions